# Identification of miR-30c-5p microRNA in Serum as a Candidate Biomarker to Diagnose Endometriosis

**DOI:** 10.3390/ijms25031853

**Published:** 2024-02-03

**Authors:** Lucía Chico-Sordo, Tamara Ruiz-Martínez, Mónica Toribio, Roberto González-Martín, Emanuela Spagnolo, Francisco Domínguez, Alicia Hernández, Juan A. García-Velasco

**Affiliations:** 1IVIRMA Global Research Alliance, IVI Foundation, Instituto de Investigación Sanitaria La Fe (IIS La Fe), 46026 Valencia, Spain; lucia.chico@ivirma.com (L.C.-S.); francisco.dominguez@ivirma.com (F.D.); juan.garcia.velasco@ivirma.com (J.A.G.-V.); 2Gynaecology Department, La Paz University Hospital, 28046 Madrid, Spain; 3IVIRMA Global Research Alliance, IVIRMA Madrid, 28023 Madrid, Spain; 4School of Health Sciences, Medical Specialties and Public Health, Obstetrics and Gynecology Area, Rey Juan Carlos University Alcorcón, 28922 Madrid, Spain

**Keywords:** endometriosis, biomarker, microRNAs, diagnosis, serum, miR-30c-5p

## Abstract

The diagnosis of endometriosis by laparoscopy is delayed until advanced stages. In recent years, microRNAs have emerged as novel biomarkers for different diseases. These molecules are small non-coding RNA sequences involved in the regulation of gene expression and can be detected in peripheral blood. Our aim was to identify candidate serum microRNAs associated with endometriosis and their role as minimally invasive biomarkers. Serum samples were obtained from 159 women, of whom 77 were diagnosed with endometriosis by laparoscopy and 82 were healthy women. First, a preliminary study identified 29 differentially expressed microRNAs between the two study groups. Next, nine of the differentially expressed microRNAs in the preliminary analysis were evaluated in a new cohort of 67 women with endometriosis and 72 healthy women. Upon validation by quantitative real-time PCR technique, the circulating level of miR-30c-5p was significantly higher in the endometriosis group compared with the healthy women group. The area under the curve value of miR-30c-5p was 0.8437, demonstrating its diagnostic potential even when serum samples registered an acceptable limit of hemolysis. Dysregulation of this microRNA was associated with molecular pathways related to cancer and neuronal processes. We concluded that miR-30c-5p is a potential minimally invasive biomarker of endometriosis, with higher expression in the group of women with endometriosis diagnosed by laparoscopy.

## 1. Introduction

Endometriosis is a chronic estrogen-dependent inflammatory disease characterized by the presence of endometrial tissue outside the uterine cavity [1]. The etiology and pathophysiology of endometriosis are unknown, but different theories have been proposed so far to explain its origin [2].

Currently, despite its high prevalence, there is a diagnostic delay estimated between 6 and 10 years from the onset of symptoms to clinical diagnosis [3,4]. The identification of precise minimally or non-invasive diagnostic methods that allow early diagnosis of the disease would improve the quality of life of these women. Different factors and molecules such as cytokines, chemokines or oxidative stress indicators have been proposed as biomarkers in the last decades [5]. However, none of them have been accurately effective in detecting endometriosis [6]. One of the most used clinical markers of endometriosis is the elevated serum level of carcinogenic antigen 125 (CA-125) [7]. This glycoprotein, frequently used in the diagnosis of ovarian cancer, when used to diagnose endometriosis increases the false positive results [8]. One of the main reasons why CA-125 fails as a diagnostic method is that its blood levels fluctuate across the menstrual cycle [9].

Recently, microRNAs (miRNAs) have emerged as potential biomarkers of disease because they are present in extracellular body fluids such as blood, saliva, urine or uterine fluid [10,11,12,13]. miRNAs are bound to protein complexes or contained in exosomes, which characterize them with a high stability in biofluids [14]. miRNAs are small non-coding RNA molecules (approximately 22 nucleotides) involved in multiple biological processes. These miRNAs play a role in regulating gene expression at a post-transcriptional level through transcriptional degradation or gene silencing [15,16]. These molecules stand out for their high stability [17]—an important criterion in the search for clinical biomarkers. In fact, the analysis of different miRNAs has been investigated as an important tool in the diagnosis, treatment and understanding of cancer [18,19,20,21], neurological disorders [22,23,24] and cardiovascular diseases [25,26,27].

The study of miRNAs as biomarkers of endometriosis has been performed in saliva [28], serum [29,30,31], plasma [32,33] and endometrial samples [34,35,36]. With the aim of identifying biomarkers of endometriosis, one of the most studied biofluids is blood because it is easy to obtain and reflects physiological and/or pathological conditions [37,38]. However, although many studies have shared the same objective and have evidenced differential expression of miRNAs in endometriosis compared with healthy women, the differences in study design and methodological issues imply that the results are not reproducible [39] and limits the potential diagnostic of miRNAs [40]. The main limitations of this type of study are the absence of quality controls in the RNA extraction and retro-transcription processes, the lack of sample hemolysis controls [31,33,41,42,43], the variability in the endogenous controls used in the normalization of miRNAs expression [44,45,46,47] and in the relative quantification method used in reverse transcription–quantitative polymerase chain reaction (RT-qPCR) [48].

Then, the main objective of this study was to evaluate the diagnostic ability of miRNAs present in serum as minimally invasive biomarkers of endometriosis. Following this objective, we identified serum miRNAs differentially expressed in endometriosis using a real-time PCR specific panel and validated a selection of these miRNAs in a novel cohort.

## 2. Results

### 2.1. Study Participant Characteristics

The characteristics of the patients are shown in Table 1, divided according to whether they were used in the preliminary study to identify differentially expressed miRNAs (*n* = 20) or in the subsequent validation of candidate miRNAs (*n* = 139). The study cohort used for the preliminary analysis (miRNA qPCR profiling) did not show statistically significant differences in age and body mass index (BMI) parameters. Regarding the cohort used in the miRNA qPCR validation, the mean age (±standard deviation) of controls and endometriosis patients was 31.7 ± 5.2 years and 37.5 ± 7.0 years, respectively. There was a significant difference in the parameter age (*p* < 0.001), making it necessary to correct the next results using this variable.

### 2.2. Identification of Serum Levels of Circulating miRNAs in Case and Control Groups

In the preliminary analysis (*n* = 20) using a serum/plasma-specific miRNAs panel, 185 different miRNAs were tested. Out of these 185 miRNAs tested in the panel, 105 miRNAs were identified in all samples, with an average of 165 miRNAs detectable per sample. When comparing the endometriosis and control groups, 29 miRNAs were found to be differentially expressed using a cutoff of *p*-value < 0.05 (Table 2, upper). The miRNAs with a higher stability (stability value ≤ 0.004) were considered candidates for miRNA normalization (Table 2, medium). The constant level of spike-in assays and the non-statistically significant differences of these controls indicated that extraction, reverse transcription and qPCR were successful (Table 2, lower).

### 2.3. Suitability of miR-30c-5p as Potential Endometriosis Biomarker

Of the 29 differentially expressed miRNAs in the serum-specific qPCR panel, 9 of them were validated in a larger population (*n* = 139). The miRNAs let-7c-5p and miR-210-3p were discarded because the quantification cycle (Cq) values were greater than 35 (detection threshold), indicating a low amount of these miRNAs in the reaction. miR-185-5p, miR-25-3p, miR-342-3p, miR-125b-5p, miR-142-3p and miR-92a-3p were not differentially expressed in women with endometriosis compared with control women (Figure 1a). However, miR-30c-5p was significantly higher in the endometriosis group (Figure 1b). No significant difference in the mean Cq values of miRNAs miR-30e-5p and miR-15b-5p was found between women with endometriosis and the control (*p* = 0.551) during the validation, so these miRNAs were used as reference miRNAs to standardize the results of the candidate miRNAs (Figure 1c). To assess the utility of miR-30c-5p in diagnosing endometriosis, we calculated the receiver operating characteristic (ROC) curve. The area under the curve (AUC) value of miR-30c-5p was 0.8437, with a sensitivity and specificity of 0.7793 and 0.9081, respectively (Figure 1d). In addition, the usefulness of miR-30c-5p as a serum biomarker of endometriosis, independently of the phase of the menstrual cycle, was confirmed by comparing its levels between different phases of the menstrual cycle in the group of healthy women (Figure A1).

### 2.4. Serum miR-30c-5p Levels in Endometriosis Stages

To evaluate whether miR-30c-5p expression correlates with the stage of endometriosis, the levels of this miRNA in different stages were analyzed. Endometriosis stages I and II could not be analyzed because the sample size in these stages was small. The analysis of the expression levels of miR-30c-5p, based on the stage of the disease according to American Society of Reproductive Medicine (ASRM) classification, showed that this miRNA could distinguish stage III (Figure 2a) and IV (Figure 2b) from the control group. However, there was no statistically significant difference between stage III and IV (Figure 2c).

### 2.5. The Impact of Hemolysis on Potential Diagnosis of miR-30c-5p

The level of hemolysis of the samples was variable, and this was assessed using the absorbance of hemoglobin at 414 nm (Appendix A) and delta Cq [(miR-23a-3p)–(miR-451a)] (Appendix A). Both methods of hemolysis assessment correlated with each other (Appendix A). However, there was no correlation between hemolysis, assessed by hemoglobin absorbance, and the miR-30c-5p value (Figure 3a). When hemolysis was measured using the delta Cq, there was a correlation between hemolysis levels and the value of the miRNA biomarker candidate (Figure 3b). After filtering for hemolysis, if delta Cq [(miR-23a-3p)–(miR-451a)] ≤ 9 the endometriosis group registered higher miR-30c-5p compared with the control group (Figure 3c), and there was no correlation with hemolysis levels in the samples (Figure 3d). The AUC value of miR-30c-5p was 0.8095 when the criterion used to monitor hemolysis was delta Cq ≤ 9 (Figure 3e).

### 2.6. Prediction of Target Genes and Functional Enrichment

A total of 1545 target genes were predicted for miR-30c-5p in the miRDB database (Appendix A). To unravel the biological impact in which miR-30c-5p may be involved, we evaluated the biological processes, cellular components and molecular functions using gene ontology and the Kyoto Encyclopedia of Genes and Genomes (KEGG) pathways related to these 1545 target genes.

After enrichment analysis, the miR-30c-5p target genes mainly belong to gene ontology biological processes related to neurogenesis and neuronal differentiation (Figure 4a, Appendix A) and gene ontology cellular components related to neurons and the endomembrane system (Figure 4a, Appendix A). Finally, gene ontology molecular functions related to the miR-30c-5p target genes include acting as kinases and DNA-binding proteins (Figure 4a, Appendix A).

Additionally, after evaluating the KEGG pathways for which these 1545 target genes are enriched, miRNAs in cancer and axon guidance pathways outstand, although KEGG pathways related to several signaling pathways and cellular processes such as autophagy and cellular senescence are also enriched (Figure 4b, Appendix A).

## 3. Discussion

Our results show the potential of miR-30c-5p as a biomarker of endometriosis in serum, maintaining its diagnostic potential even when there is an acceptable risk of hemolysis in blood samples.

The gold standard technique for the diagnosis of endometriosis is laparoscopy [50]. Despite its high diagnostic sensitivity and specificity, this technique is highly invasive and depends on the finding of lesions identifiable by the surgeon. In the last few years, the relentless search for a reliable and accurate diagnostic test for the early diagnosis of endometriosis has focused on the role of miRNAs. Despite the large number of studies on miRNAs as biomarkers, no validation studies have subsequently been carried out. A differential expression of miRNAs between women with endometriosis and healthy women has been evidenced in both plasma [32,33] and serum samples [29,30,31]. However, serum has certain advantages over plasma for several reasons. Firstly, the probability of contamination by platelets and erythrocytes is lower [51]. Secondly, the interference of the compounds used for serum collection with subsequent processes to quantify miRNAs such as reverse transcription and/or PCR is acceptable compared with the interference of those used for plasma collection [52]. In addition, in the case of performing a validation study from the clinic’s sample repository, it is easier to perform it on serum as there is more availability [53].

There is great variability in the miRNAs reported as candidate biomarkers of endometriosis despite using the same type of sample (serum) and the same technique (RT-qPCR) for the quantification of miRNAs. Mainly, the difference is in the endogenous control used for data normalization [54] and in the applied equation [55]. A large number of studies use U6 small nuclear RNA (RNU6) to normalize the expression of miRNAs in serum [29,30,31,47,56,57,58]. However, its use in normalizing data for circulating serum cell-free miRNAs is not recommended because RNU6 has been shown to have high variability between samples [59,60]. In our study, the following two different methods were used for normalization of miRNA expression levels: normalization based on the overall mean of miRNAs detected in all samples and normalization using two reference miRNAs’ mean with stable expression in both study groups. First, according to the results of the study by Mestdagh and coworkers, for the identification of differentially expressed miRNA candidates from a panel of 185 miRNAs, the mean of all expressed miRNAs was used as a normalization strategy [44]. Secondly, once the candidate miRNAs were identified in a discovery cohort, some of them were validated in a new cohort of patients; in this case, the mean of miR-30e-5p and miR-15b-5p was used for normalization, whose expression was equal in the control and endometriosis groups. The use of two or more reference miRNAs for normalization has been used in several studies in plasma [33,61,62], and their use is highly recommended in the Minimum Information for Publication of Quantitative Real-time PCR Experiments (MIQE) [55].

Preliminary analysis of a panel of 185 miRNAs with subjects from each of the groups allowed us to obtain not only endogenous miRNAs suitable for our study but also a list of differentially expressed miRNAs in the endometriosis group. From this panel, miRNA biomarker candidates were selected and quantified by RT-qPCR. Our results evidenced that not all differentially expressed miRNAs during the profiling maintain their diagnostic potential after validation in a new cohort, so validation is recommended. Consequently, it is assumed that there will be more miRNAs not included in this panel of 185 miRNAs that could be putative biomarkers for endometriosis. In our study, this specific serum/plasma panel was chosen for preliminary analysis because most of the miRNA biomarker candidates for endometriosis suggested in the previous literature were included. However, there are other alternatives for preliminary analysis such as next-generation sequencing (NGS), which allows global miRNA screening, followed by subsequent validation by RT-qPCR, as in our study [33]. Further, miR-30c-5p would be a good cell-free serum biomarker for endometriosis because we found a statistically significant difference in its levels between women with endometriosis and control women, and it registered high sensitivity and specificity values. Previously, studies reported no difference in the expression levels of candidate miRNA biomarkers for endometriosis based on the phase of menstrual cycle in the control group [31,56]. Our biomarker candidate, miR-30c-5p, is unaltered by menstrual cycle phase when comparing the proliferative and secretory phase (even if we include subjects whose phase is unknown) in healthy women. The study by Wang and coworkers (2016) evidenced lower levels of miR-30c-5p in the serum of patients with endometriosis, whereas we observed the opposite trend. This difference could be due to either the method used in normalization, as our study used the mean of two stable endogenous miRNAs present in the samples versus their use of an exogenous synthetic miRNA (cel-miR-39) introduced prior to RNA extraction. Another explanation for our discrepancies could be that all endometriosis patients included were diagnosed at stage I/II [46]. Other studies have also reported lower expression levels of this miRNA in ectopic [63] and eutopic endometrium tissue samples [64], and decreased proliferation, adhesion, invasion and migration of endometrial stromal cells have been observed following miR-30c-5p upregulation [64]. However, in both studies, they used RNU6 as the internal reference gene for normalization. Furthermore, it is of great importance to specify which mature transcript was quantified, 5p or 3p [65,66]. Based on the sequence of the primer used, Chen and coworkers (2017) evaluated miR-30c-5p, while Zhang and coworkers (2022) quantified the miRNA precursor [63,64]. Both the precursor and mature miRNA can be quantified by RT-qPCR, although the levels may be different because the precursor undergoes faster degradation than the mature one [67].

In all other candidate miRNAs tested in the validation cohort, we found no statistically significant differences between the endometriosis and control groups, although these differences were detected in the profiling analysis. Previous studies have analyzed the potential of some of the proposed miRNAs as biomarkers of endometriosis, or as possible therapeutic targets. miR-342-3p and miR-125-5p were detected overexpressed in the serum of women with endometriosis [31,56]. Studies in plasma samples suggested miR-185-5p [68], miR-125b-5p [62] and miR-92a-3p [69,70] as good biomarkers of endometriosis. Other studies analyzed the effect of under- or overexpression of miR-185-5p [71], miR-142-3p [72,73] and miR-92a-3p [74] on the phenotype of eutopic and/or ectopic endometrial stromal cells, altering their properties and positioning them as future therapeutic alternatives. Another of the miRNAs recurrently suggested as an endometriosis biomarker is miR-451a [31,56,57]. This miRNA is highly expressed in erythrocytes; for this reason, it is used to test for the presence of hemolysis in serum and plasma samples [75,76,77,78,79]. Hemolysis is a major source of variation in these types of samples as it involves contamination with miRNAs from cells [41]. Although many hemolytic diseases exhibit increased levels of hemolysis in blood samples [80,81,82], the diagnostic potential of miR-451a in endometriosis is questionable because it could act as a confounding variable [83,84]. The levels of miR-451a and miR-23a-3p allowed us to calculate the delta Cq [(miR-23a-3p)–(miR-451a)] to assess the risk of hemolysis as an alternative to hemoglobin absorbance [41,85]. In our study, we analyzed the impact of hemolysis on the diagnostic potential of our candidate miRNA, miR-30c-5p. For this purpose, the hemolysis level of the samples from both study groups was similar, evaluated with both hemoglobin absorbance and miRNA delta Cq, which showed a positive correlation with each other. When we analyzed whether there was a correlation between the risk of hemolysis and the level of miR-30c-5p, its levels only showed a statistically significant correlation with the hemolysis delta Cq. In order to obtain hemolysis-free serum samples, certain precautions need to be taken during the extraction process [86]. In addition, there is a possibility that a sample first classified as non-hemolyzed may have hemolysis after measuring the hemoglobin absorbance or delta Cq [85]. This would involve obtaining a new sample, with the inconvenience for the attending patient [87]. However, when only samples whose delta Cq ≤ 9 were taken into account, miR-30c-5p maintained differential expression between the control and endometriosis groups. Moreover, the variable hemolysis did not correlate with candidate miRNA levels, preserving its diagnostic potential.

Analysis of the molecular pathways in which miR-30c-5p is implicated according to the databases highlighted pathways related to cancer and neuronal processes. Previously, differential expression of miRNAs whose target genes were enriched in cancer-related pathways has been reported in patients with endometriosis [88]. In fact, studies on different types of cancer such as hepatocellular carcinoma [89], oral squamous cell carcinoma [90], gastric cancer [91], papillary thyroid carcinoma [92] or bladder cancer [93] have evidenced a lower expression of this miRNA in neoplastic tissues. In our study, women suffering from endometriosis showed increased serum miR-30c-5p expression. Molecular links exist between cancer and endometriosis, because the main cells involved in both pathologies are able to evade the apoptosis process, share characteristics with stem cells regarding division capacity and standout for a high angiogenic potential [94]. Endometriosis is a benign gynecological condition, although there are studies that relate it to an increased risk of some types of cancer such as ovarian cancer, breast cancer or endometrial cancer [95]. Studies on ovarian [96] or breast cancer [97], also estrogen-dependent conditions like endometriosis, showed that increased levels of miR-30c-5p inhibit tumor cell proliferation, migration and invasion. Therefore, the fact that this miRNA is elevated in patients with endometriosis as opposed to patients with ovarian or breast cancer could underline the differences between these two separate conditions. However, more studies are needed to further investigate the molecular mechanisms in which miR-30c-5p is involved in both endometriosis and different types of cancer to clarify its role in each of them. One of the potential molecular pathways that could be key in the role of miR-30c-5p in endometriosis and cancer is related to the TP53 (p53) gene. The presence of mutations in TP53, and consequently the elicitation of mutant p53 proteins, characterizes a large proportion of human tumors [98,99]. Studies in cancer have evidenced that there is a relationship between p53 and miR-30c, concluding that the reduced expression of miR-30c correlates with increased mutations in p53, which in turn increases tumor aggressiveness [100]. These results suggest that differences in miR-30c-5p levels, and consequently in molecular pathways regulated by this miRNA, could be associated with the activation or not of the cellular malignization process, which differentiates endometriosis from cancer [96,97,101,102,103]. Also, this miRNA could be used as a differential diagnostic tool between both pathologies.

Other miR-30c-5p-associated pathways are related to the processes of neuron formation and differentiation and axonal conduction. Recent studies suggested that endometriosis affects the central nervous system [104,105]. The unifying thread that links neural processes and endometriosis is the chronic pain experienced by patients [106,107]. This endometriosis-associated pain is considered a type of neuropathic pain [108,109,110,111]. In fact, one of the most studied genes has been the brain-derived neurotrophic factor (BDNF) [112], which has even been suggested as a neuronal biomarker for endometriosis diagnosis [113]. Tramullas and coworkers in their studies demonstrated a causal relationship of miR-30c-5p levels with nociception and neuropathic pain in rodents [114,115]. Furthermore, they suggested that this relationship was mediated by transforming growth factor-beta 1 (TGF-β1) [114], which was increased in endometriosis and was related to an increased migratory, invasive and colonizing potential of endometriotic cells [116].

Our study stands out for several strengths based on the normalization of the miRNA levels, the inclusion of quality controls in the methodology, the quantification of the variable hemolysis and the use of different cohorts for discovery and validation. Thanks to the preliminary analysis where numerous miRNAs were tested in a small population, both candidate disease biomarker miRNAs and endogenous miRNAs could be extracted. The method used for normalization was based on the previously published literature. The incorporation of spike-in markers in both RNA extraction, retrotranscription and quantification were helpful in monitoring the technique’s efficiency. The variable hemolysis was measured in two different ways, and its impact on the miRNA biomarker candidate was analyzed. Furthermore, serum levels of miR-30c-5p showed no differences between the phases of the menstrual cycle. Therefore, the use of miR-30c-5p as a mild or severe grade of endometriosis biomarker would be a reproducible technique in other laboratories, provided that its levels are normalized with endogenous miRNAs miR-30e-5p and miR-15b-5p, and the level of serum hemolysis measured by delta Cq [(miR-23a-3p)–(miR-451a)] does not exceed a value of 9. However, one of the main limitations of this study was that no cases of women with stage I endometriosis were included, and there were very few cases with stage II.

## 4. Materials and Methods

### 4.1. Study Population and Design

A total of 159 women, from La Paz University Hospital (LPUH) and IVIRMA Madrid, participated in this study, aged 22–55 years and with a BMI between 18.5 and 29.9 kg/m^2^. Recruitment was carried out between March 2018 and January 2023. This study included women diagnosed with endometriosis after undergoing laparoscopic surgical exploration of the abdominal cavity and anatomopathological study of the lesions. The endometriosis stage was classified according to the ASRM revised classification. Women with endometriosis stages I, II, III and IV were recruited for the study. Patients treated with gonadotropin releasing hormone (GnRH) agonist or aromatase inhibitors in the last 3 months were excluded. The control group consisted of patients undergoing laparoscopic tubal ligation for contraception with the absence of endometriotic lesions from LPUH and healthy oocyte donor women from IVIRMA Madrid. These women had proven fertility at the time of sample collection as an inclusion criterion. Proven fertility was defined as at least one live birth and no history of non-progressive gestation. The exclusion criterion for the control group was the finding of endometriosis during laparoscopy. Women, from both study groups, were recruited at all phases of the menstrual cycle as this variable does not influence the serum levels of circulating miRNAs [56,57]. All patients provided written informed consent before being included in the study. Puerta de Hierro University Hospital and LPUH board (Madrid, Spain) approved the study protocol (1707-MAD-083-JG, approved in November 2017).

A preliminary real-time PCR panel analysis was performed in serum samples from women with confirmed endometriosis (n = 10) and women without endometriosis from the control group (n = 10) (Figure A2). A total of 185 miRNAs were evaluated in this miRNA qPCR serum profiling panel. From this preliminary analysis, 9 miRNAs with differential expression between the control and endometriosis groups and 2 serum miRNAs with stable expression in both groups to standardize the results were chosen for validation analysis. The miRNAs selected as biomarker candidates were validated by qPCR in a new cohort of women with endometriosis (n = 67) and women in the control group (n = 72) in two consecutive batches. The level of differentially expressed miRNAs in the validation was compared between the endometriosis stages, and the impact of hemolysis on their diagnostic potential was studied.

### 4.2. Serum Sample Collection

Blood samples (5 mL) were collected in sterile serum separator gel tubes (Sarstedt, Nümbrecht, Germany) by direct venipuncture prior to the procedure (laparoscopy or follicular puncture). The samples were immediately centrifuged at 3000× *g* for 10 min at room temperature. The serum was transferred to 2 mL tubes and stored at −80 °C until analysis.

### 4.3. miRNA qPCR Profiling

Serum samples from women with endometriosis (n = 10) and from the control group (n = 10) were analyzed using the miRNA Ready-to-Use PCR. A total of 185 miRNAs were evaluated by qPCR in serum using the miRCURY LNA SYBR Green master mix (QIAGEN, Hilden, Germany). Amplification was performed with a LightCycler^®^ 480 real-time PCR system (Roche, Mannheim, Germany) in 384-well plates. Amplification curves were analyzed using Roche LC 480 software for both Cq value determination and melting curve analysis. Normalization was performed based on the average of the assays detected in all samples (105 miRNAs), according to the following formula: normalized Cq (dCq) = global mean Cq − assay Cq (miRNA of interest). A higher value thus indicated that the miRNA was more abundant in the sample. The 2nd derivate method was used to analyze the relative changes in gene expression from expression profiling, obtaining the fold change.

### 4.4. RNA Extraction and cDNA Synthesis

The serum was thawed on ice and centrifuged at 3000× *g* for 5 min at 4 °C. Total RNA was extracted from 200 µL of serum using miRNeasy serum/plasma Advanced Kit (QIAGEN, Hilden, Germany) according to the manufacturer’s instructions. In the first extraction step, RNA spike-in template mixture (UniSp2, UniSp4 and UniSp5) was added as an extraction control. RNA was eluted in 20 µL nuclease-free water. An amount of 7 µL of RNA was retrotranscribed using the First-Strand cDNA Synthesis kit (QIAGEN, Hilden, Germany) according to the manufacturer’s guide on a Mastercycler^®^ X50a (Eppendorf, Hamburg, Germany). To confirm that reverse transcription and amplification had occurred with equal efficiency in all samples, UniSp6 RNA spike-in template was added.

### 4.5. Selection of miRNA Biomarker Candidates of Endometriosis and Reference miRNAs

From the miRNAs analyzed in miRNA qPCR profiling and also based on the previous literature, miRNA biomarker candidates of endometriosis and reference miRNAs were selected. Then, nine differentially expressed miRNAs were selected as putative biomarkers (miR-185-5p, let-7c-5p, miR-30c-5p, miR-210-3p, miR-25-3p, miR-342-3p, miR-125b-5p, miR-142-3p and miR-92a-3p) and two miRNAs with stable expression in the control and endometriosis groups were selected as reference miRNAs (miR-30e-5p and miR-15b-5p). Endogenous miRNAs were chosen based on the stability value of miRNAs found in all samples. This value was calculated using NormFinder [117] software.

### 4.6. miRNA Real-Time qPCR Validation

The 9 miRNA biomarker candidates were validated in a larger number of patients in the endometriosis (n = 67) and the control group (n = 72). This experiment was performed in two consecutive rounds; for this reason, the sample size for the miRNAs evaluated was different. In the first round, miR-185-5p, let-7c-5p, miR-30c-5p, miR-210-3p, miR-25-3p, miR-342-3p and miR-125b-5p were analyzed. In the second round, miR-30c-5p, miR-142-3p and miR-92a-3p were analyzed, discarding miRNAs without differential expression of the first round. The miRNAs were quantified by real-time qPCR on a 7500 Fast System (Applied Biosystems) in 96-well plates using miRCURY LNA SYBR Green (QIAGEN, Hilden, Germany), according to the manufacturer’s instructions. Primers (Table A1) for each miRNA were obtained from Qiagen. The Cq values of miRNAs in each subject were measured in duplicate, and the mean Cq value obtained was used in subsequent analyses. Quantification cycle values greater than 35 were considered to be below the detection level of the assay, and a UniSp3 interplate calibrator was used to calculate the calibration factor (mean plate UniSp3 Cq—mean global UniSp3 Cq). The miRNA Cq values were normalized, after interplate calibration, to the mean Cq values of two reference miRNAs. Relative expression for each miRNA in each sample was obtained using 2^−dCq^ (2^−(Cq miRNA of interest − Cq reference miRNAs)^). 

### 4.7. Hemolysis Control

Hemolysis of serum samples was monitored by the following two methods: spectrophotometry and miRNA expression. By spectrophotometry, the absorbance of oxy-hemoglobin was measured at ʎ = 414 nm using the NanoPhotometer N60 (Implent). The other method used to identify sample hemolysis risk was based on the following two miRNAs: miR-23a-3p and miR-451a. Levels of miR-23a-3p are stable in serum and are not affected by hemolysis. However, miR-451a is enriched in erythrocytes and its levels are increased in serum when hemolysis occurs. During RT-qPCR, both miRNAs were measured in duplicate in each sample, and the delta Cq [(miR-23a-3p)–(miR-451a)] was calculated (mean miR-23a-3p Cq–mean miR-451 Cq).

### 4.8. miRNA Target Gene Prediction and Functional Enrichment Analysis

Target genes that could be under the modulation of the differentially expressed miRNA at validation, miR-30c-5p, were predicted employing the miRDB database [118,119]. Subsequently, functional annotation and graphical representation of the gene ontology biological processes, cellular component and molecular functions and KEGG pathways enriched for the previously predicted target genes were performed using the ShinyGO 0.77 tool [120] and the Bioinformatics (www.bioinformatics.com.cn, accessed on 29 January 2024) free online platform.

### 4.9. Statistical Analysis

Data were represented as mean ± standard deviation (SD) for age, and BMI of study participants and statistical analysis was performed with R software (version 4.0.3). For normalization of miRNA expression levels, the overall mean of all detected miRNAs (miRNA qPCR profiling) or the mean of two endogenous miRNAs (miRNA real-time qPCR validation) was used. In the preliminary analysis, miRNA expression was calculated as the 2nd derivative. During the validation phase, relative miRNA expression was calculated with the comparative Cq method based on Schmittgen and Livak (2008) [48]. The Student *t*-test was used to compare the study variables in the endometriosis and control groups, using a cutoff of *p*-value < 0.05. miRNAs with *p*-value < 0.05 were considered differentially expressed between the endometriosis and control groups. A multivariate model was applied with the variable age in the differentially expressed miRNAs. The AUC was calculated with the ROC curve for differentially expressed miRNAs between the endometriosis and control group. The ROC curves and AUCs were calculated with R software by adding the multivariate model with age. Pearson’s correlation coefficient was used to examine the relationship between continuous variables. The variation of miR-30c-5p levels between the phases of the menstrual cycle was evaluated by ordinary one-way ANOVA test (when cases with unknown phase were taken into account) or by Student *t*-test (proliferative vs. secretory).

To assess the impact of hemolysis on the potential of miRNAs as biomarkers, candidate miRNA levels were evaluated between the endometriosis and control groups by Student *t*-test. This analysis was performed repeatedly, filtering by different delta Cq [(miR-23a-3p)–(miR-451a)] values.

## 5. Conclusions

In conclusion, serum levels of miRNAs are promising biomarkers of diseases such as endometriosis. Specifically, miR-30c-5p is postulated as a good candidate for a minimally invasive biomarker of endometriosis, allowing differentiation between healthy women and women with endometriosis in mild or advanced stages. However, to guarantee its diagnostic potential in serum, it is necessary to control the serum hemolysis. Future large-scale multicenter studies for the validation of the levels of this miRNA including women with both stage I and stage II endometriosis would be necessary. In addition, these studies could analyze the molecular pathways in which miR-30c-5p intervenes and its relationship with endometriosis, including other study groups such as women with different types of gynecological cancer, and in which the pain variable is quantified.

## Figures and Tables

**Figure 1 ijms-25-01853-f001:**
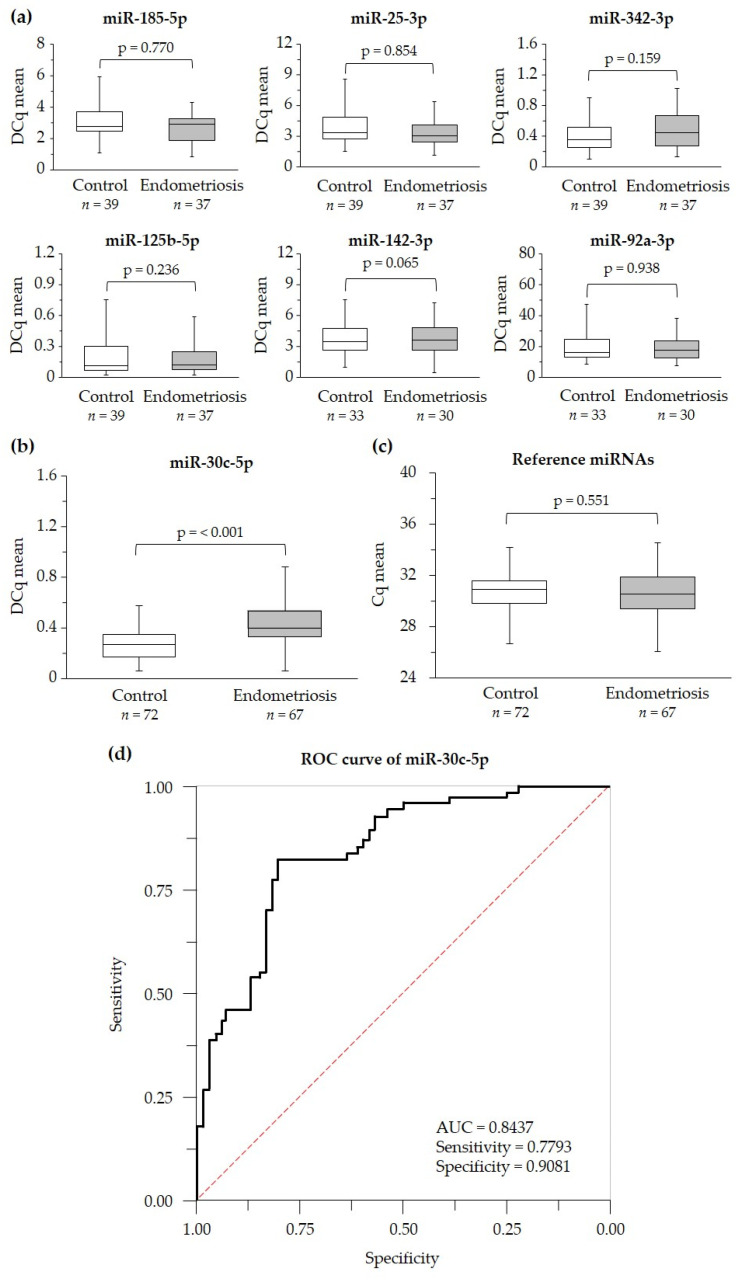
Serum expression of miRNAs in control and endometriosis groups in the validation cohort. (**a**) Expression of miR-185-5p, miR-25-3p, miR-342-3p, miR-125b-5p, miR-142-3p and miR-92a-3p in the control and endometriosis groups. (**b**) Differential expression of miR-30c-5p between the control and endometriosis groups. (**c**) Mean value of the quantification cycle of the miRNAs chosen as endogenous (miR-30e-5p and miR-15b-5p) used in the normalization of the expression levels of candidate biomarker miRNAs. (**d**) Receiver operating characteristic curve (ROC) of serum miR-30c-5p obtained after adding the multivariate model with age. The corresponding area under the curve (AUC), sensitivity and specificity are indicated. Box plots were used to represent the levels of each miRNA (**a**,**b**) or the Cq mean (**c**). Data were plotted showing median and interquartile range. Results were compared using Student’s *t*-test and were considered statistically significant when *p* < 0.05. *n* indicates the number of individuals analyzed. DCq indicates miRNA Cq values normalized to the mean Cq values of reference miRNAs. Cq indicates the quantification cycle of miRNA detection by quantitative PCR.

**Figure 2 ijms-25-01853-f002:**
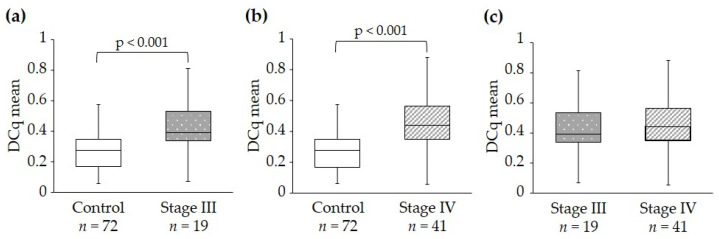
miR-30c-5p expression in different stages of endometriosis. (**a**) Expression of miR-30c-5p in the control group and patients with stage III endometriosis. (**b**) Comparison of miR-30c-5p expression between the control group and the stage IV endometriosis group. (**c**) Expression of miR-30c-5p in endometriosis subjects divided into stage III and IV. Endometriosis stage assignment was based on the ASRM classification. Box plots were used to represent miR-30c-5p levels after normalization with reference miRNAs. Data were plotted showing median and interquartile range. Results were compared using Student’s *t*-test and were considered statistically significant when *p* < 0.05. *n* indicates the number of individuals analyzed. DCq indicates miRNA Cq values normalized to the mean Cq values of reference miRNAs.

**Figure 3 ijms-25-01853-f003:**
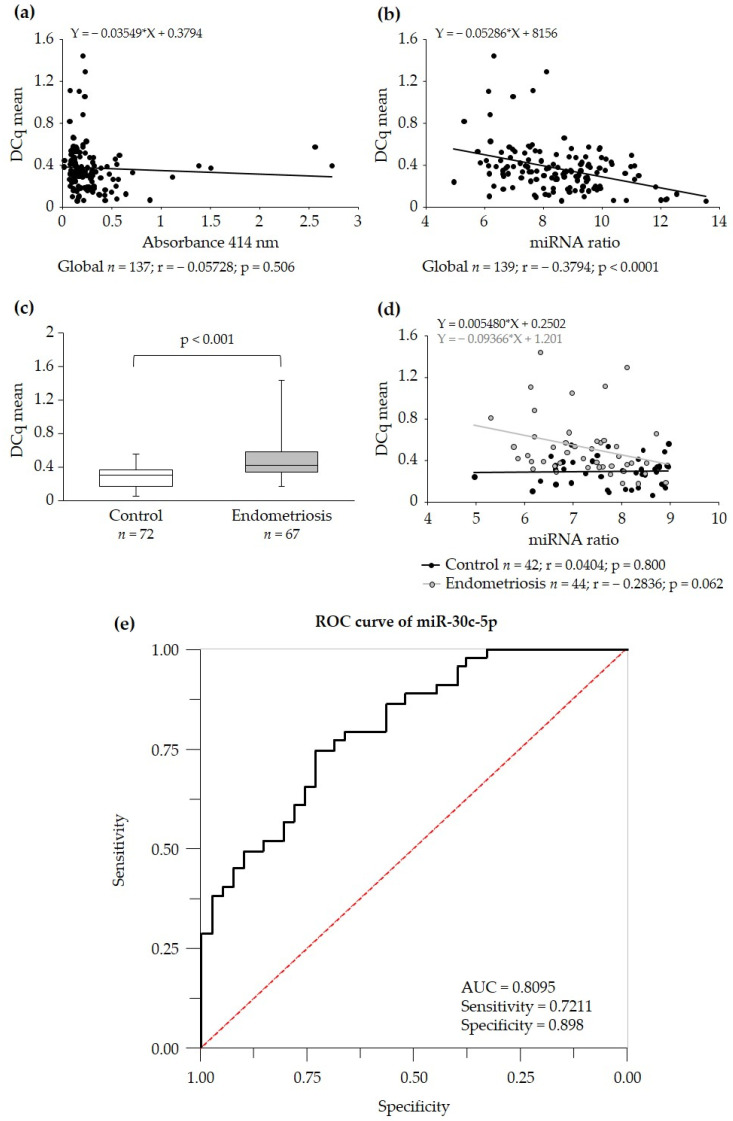
Variation of miR-30c-5p levels with the level of hemolysis. (**a**) Global correlation between hemoglobin absorbance and miR-30c-5p levels. (**b**) Global correlation between the delta Cq [(miR-23a-3p)–(miR-451a)] and miR-30c-5p levels. (**c**) Box plot of miR-30c-5p levels between the control and endometriosis groups, including samples with a delta Cq ≤ 9, excluding from the analysis those with a higher delta Cq. (**d**) Correlation between the delta Cq [(miR-23a-3p)–(miR-451a)] and miR-30c-5p levels in samples with a ratio ≤ 9, divided into controls (black) and women with endometriosis (grey). (**e**) Receiver operating characteristic curve (ROC) of serum miR-30c-5p when hemolysis delta Cq ≤ 9. The curve was obtained after adding the multivariate model with age. The corresponding area under the curve (AUC), sensitivity and specificity are indicated. r represents Pearson’s correlation coefficient, and the equation of the line is indicated (a, b and d). Statistically significant data when *p* > 0.05. *n* represents the number of individuals analyzed. DCq indicates miRNA Cq values normalized to the mean Cq values of reference miRNAs.

**Figure 4 ijms-25-01853-f004:**
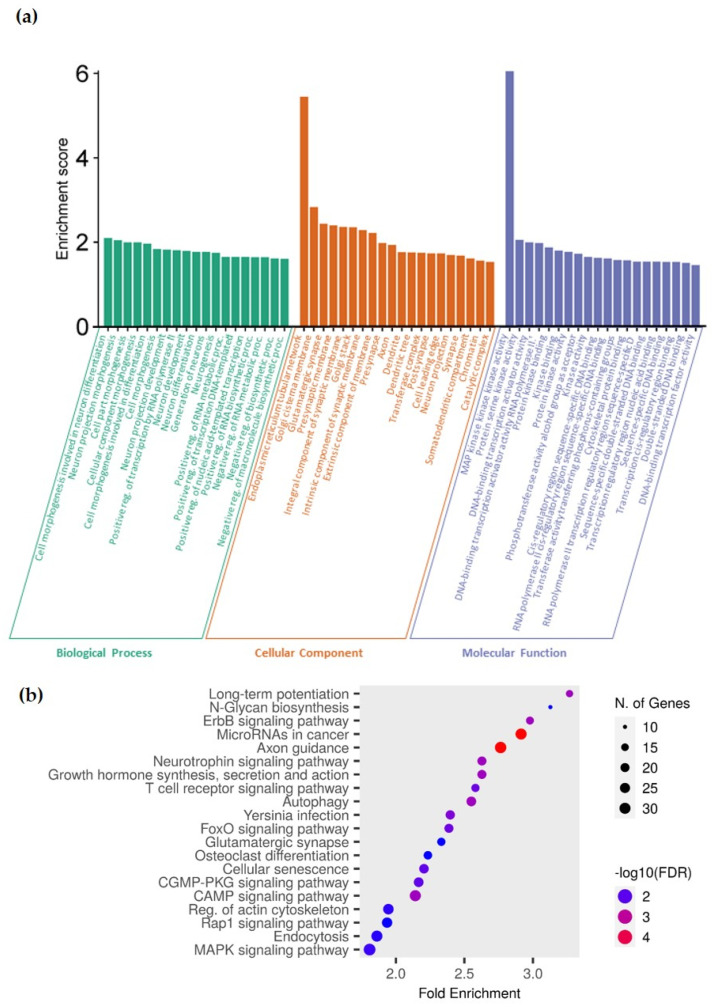
Functional enrichment of hsa-miR-30c-5p target genes. (**a**) Gene ontology function analysis histogram of miR-30c-5p target genes. Dark cyan represents top 20 gene ontology biological processes, sienna represents top 20 gene ontology cellular component and steel blue represents top 20 gene ontology molecular function. (**b**) Dot plot of KEGG pathways enriched in miR-30c-5p target genes. Fold enrichment represents the overrepresentation of genes from certain KEGG pathways, dot size is directly proportional to the number of genes involved in the pathway and color gradient represents the FDR.

**Table 1 ijms-25-01853-t001:** Demographics and clinical characteristics of study participants.

	miRNA qPCR Profiling *n* = 20		miRNA Real-Time qPCR Validation *n* = 139	
Variable	Control Group *n* = 10	Endometriosis Group *n* = 10	Control Group *n* = 72	Endometriosis Group *n* = 67
Age ± SD (years)	32 ± 4.6	34.9 ± 4.0	31.7 ± 5.2	37.5 ± 7.0
Body Mass Index ± SD (kg/m^2^)	22.5 ± 2.4	22.6 ± 3.2	24.1 ± 3.0	22.8 ± 3.1
ASRM stage, *n* (%)				
I	NA	0 (0)	NA	1 (1.5)
II	NA	4 (40)	NA	6 (9.0)
III	NA	4 (40)	NA	19 (28.3)
IV	NA	2 (20)	NA	41 (61.2)
Phase of menstrual cycle, *n* (%)				
Proliferative	10 (100)	0	49 (68.1)	0
Secretory	0	0	7 (9.7)	0
Unknown	0	10 (100)	16 (22.2)	67 (100)
Pain symptoms, *n* (%)				
Any pain symptoms	10 (100)	2 (20)	72 (100)	6 (8.9)
Dyspareunia	0	3 (30)	0	30 (44.7)
Dysmenorrhea	0	7 (70)	0	51 (76.1)
Dyschezia	0	3 (30)	0	22 (32.8)
Dysuria	0	1 (10)	0	9 (13.4)
Chronic pelvic pain	0	4 (40)	0	42 (62.6)
Infertility, *n* (%)				
Primary	0	3 (30)	0	28 (41.7)
Secondary	0	1 (10)	0	1 (1.4)
Unknown	0	5 (50)	0	23 (34.3)
Proven fertility (live birth)	10 (100)	1 (10)	72 (100)	15 (22.3)
Tumoral makers ± SD				
CA-125 (U/mL)	NA	32.4 ± 17.4	NA	185.3 ± 527.8
CA-19.9 (U/mL)	NA	28.1 ± 19.2	NA	94.99 ± 268.2
HE4 (pmol/l)	NA	-	NA	76.43 ± 31.05
Comorbidities, *n* (%)				
Hypothyroidism	0	0	2 (2.7)	11 (16.4)
Asthma	0	0	2 (2.7)	7 (10.4)
Migraine	0	0	1 (1.3)	4 (5.9)
Allergic rhinoconjunctivitis	0	0	1 (1.3)	3 (4.4)
Vitiligo	0	0	0	3 (4.4)
Hypertension	0	0	0	3 (4.4)
Depression	0	0	0	3 (4.4)
Fibromyalgia	0	0	0	2 (2.9)
Raynaud Syndrome	0	0	0	2 (2.9)
Other	0	1 (10)	4 (5.5)	8 (11.9)

Data are represented as mean ± standard deviation. NA—not applicable; ASRM—American Society for Reproductive Medicine. There is no tumor marker HE4 value for the endometriosis group used in miRNA qPCR profiling because only 1 of the 10 patients recorded this test result. The comorbidities with the highest frequency in our study cohort are listed. The “Other” comorbidities section includes inflammatory traits (celiac disease, Crohn’s disease and multiple sclerosis), gastrointestinal features (chronic gastritis) and mental disorders (anxiety, ADHD, bipolar and personality disorders) based on McGrath et al.’s (2023) review [49].

**Table 2 ijms-25-01853-t002:** Differentially expressed miRNAs in serum, reference miRNA candidates and technical controls in qPCR profiling preliminary study.

Differentially Expressed miRNAs	Assay Catalog No.	*p*-Value	Fold Change
[miR-210-3p] *	YP00204333	0.005	−1.73
[miR-185-5p] *	YP00206037	0.006	−1.44
[miR-25-3p] *	YP00204361	0.007	−1.44
miR-660-5p *	YP00205911	0.007	−1.44
[miR-142-3p]	YP00204291	0.008	1.55
[miR-92a-3p] *	YP00204258	0.010	−1.37
miR-28-5p	YP00204322	0.010	1.85
let-7a-5p	YP00205727	0.010	1.35
miR-532-5p *	YP00204003	0.014	−1.43
miR-16-2-3p *	YP00204309	0.014	−1.43
[let-7c-5p]	YP00204767	0.015	1.41
miR-16-5p *	YO00205702	0.017	−1.44
miR-766-3p	YP00204499	0.019	1.73
miR-451a *	YP02119305	0.020	−1.70
miR-7-1-3p	YP00205888	0.023	2.24
miR-19b-3p *	YP00204450	0.023	−1.29
miR-132-3p	YP00206035	0.027	1.57
miR-192-5p *	YP00204099	0.029	−1.43
miR-215-5p *	YP00204598	0.030	−1.51
[miR-30c-5p]	YP00204783	0.031	1.33
let-7g-5p	YP00204565	0.031	1.13
miR-338-3p	YP00204719	0.034	1.58
miR-19a-3p *	YP00205862	0.034	−1.25
miR-140-3p *	YP00204304	0.037	−1.26
miR-30b-5p	YP00204765	0.039	1.29
miR-141-3p *	YP00204504	0.040	−1.73
let-7d-5p	YP00204124	0.046	1.36
[miR-342-3p] *	YP00205625	0.046	−1.49
miR-150-5p *	YP00204660	0.048	−1.69
**Reference miRNA candidates**	**Assay catalog no.**	**Stability value**	
miR-30e-5p	YP00204714	0.003	
miR-652-3p	YP00204387	0.003	
miR-29b-3p	YP00204679	0.004	
miR-21-5p	YP00204230	0.004	
miR-15b-5p	YP00204243	0.004	
miR-22-3p	YP00204606	0.004	
miR-107	YP00204468	0.004	
**Technical controls (spike-ins)**	**Assay catalog no.**	***p*-value**	
*RNA isolation control*			
UniSp2	YP00203950	0.796	
UniSp4	YP00203953	0.971	
UniSp5	YP00203955	0.218	
*cDNA synthesis control*			
UniSp6	YP00203954	0.684	
*Interplate calibrator*			
UniSp3	YP02119288	0.143	

* miRNAs down-regulated in the endometriosis group. [] indicates the miRNAs selected for further validation as endometriosis candidates in a new cohort, with the exception of miR-125b-5p, which was selected from the literature. Fold change calculated by 2nd derivate method.

## Data Availability

The data supporting the findings of this study are available on request from the corresponding author.

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
