# Peer review of "Identification of miR-30c-5p microRNA in Serum as a Candidate Biomarker to Diagnose Endometriosis"

_ijms, 2024, doi:10.3390/ijms25031853_

Round 1

Reviewer 1 Report

Comments and Suggestions for Authors

The paper entitled “Identification of miR-30c-5p microRNA in serum as a candidatebiomarker to diagnose endometriosis” by Chico-Sordo et al. identify candidate serum microRNAs associated with endometriosis to be used as biomarker. The topic of this work is for sure interesting, and in the recent literature there are various paper related to miR 30c-5p and on endometriosis. The English form is fine and the overall quality of the paper is good to be published in MDPI IJMS. I only have a few points that should be adjusted by the authors before publication: - all the acronyms must be introduced in the extended form the first time they appear in the text. This is not in your text. For instance, “AUC” in the abstract, or “BMI” in the paragraph 2.1 are explained different pages after into section 4. Please, revise the whole text to give the readers all the information required to understand the text. - The same for the KEGG pathways. I’m not sure that everyone knows what this means. - Figure 4 is not really clear to me. Is it possible to use a different plot or to explain it better into the text? - Do you have any idea of why this miR-30c-5p is overexpressed in endometriosis while is downregulated in some cancer diseases, such as breast cancer? Can you please add some more details into the discussion, and not only “These results suggest that variation in miR-30c-5p levels could have a role in the process of cell mutation and malignancy, a fact that differentiates endometriosis from cancer”

Author Response

January 30, 2024

Dear Reviewer:

Thank you very much for taking the time to review our article entitled "Identification of miR-30c-5p microRNA in serum as a candidate biomarker to diagnose endometriosis". We appreciate the comments and suggestions of reviewers, as they have allowed us to improve the article and give more weight, credibility and depth to our results. Please find the detailed responses below and the corresponding revisions/corrections highlighted/in track changes in the re-submitted files. The line numbers detailed in the responses to the reviewers correspond to the manuscript titled "ijms-2845933-revised" where the modifications are marked. The final clean manuscript has also been uploaded to the platform.

Comment 1:

All the acronyms must be introduced in the extended form the first time they appear in the text. This is not in your text. For instance, “AUC” in the abstract, or “BMI” in the paragraph 2.1 are explained different pages after into section 4. Please, revise the whole text to give the readers all the information required to understand the text.

Response 1:

We agree. We have, accordingly, revised the whole text to define each abbreviation the first time it is mentioned.

See: Page 1, Abstract section, lines 23-24; Page 2, Results section, lines 82, 83; Page 5, Results section, lines 109, 118, 119, 130; Page 7, Results section, line 175; Page 13, Materials and methods section, lines 384, 388-389; Page 14, Materials and methods section, lines 425-426; Page 15, Materials and methods section, lines 488, 503-504.

Comment 2:

The same for the KEGG pathways. I’m not sure that everyone knows what this means.

Response 2:

We agree. We have added the description “Kyoto Encyclopedia of Genes and Genomes” the first time that the term appeared.

See the same pages indicated in the Response 1.

Comment 3:

Figure 4 is not really clear to me. Is it possible to use a different plot or to explain it better into the text?

Response 3:

Although we consider both functional enrichment maps and dot plots to be the usual, clearest and most representative ways of presenting these results, we have replaced the functional enrichment map for gene ontology biological processes with a histogram showing the top 20 biological processes, cellular components and molecular functions related to the 1545 miR-30c-5p target genes. A more detailed explanation has also been incorporated in the text.

See: Page 7, Results section, lines 173-176, 177-182; Page 9, Figure 4 and lines 200-206; Page 15, Materials and methods section, lines 486-487, 489-490.

Comment 4:

Do you have any idea of why this miR-30c-5p is overexpressed in endometriosis while is downregulated in some cancer diseases, such as breast cancer? Can you please add some more details into the discussion, and not only “These results suggest that variation in miR-30c-5p levels could have a role in the process of cell mutation and malignancy, a fact that differentiates endometriosis from cancer”

Response 4:

Indeed, in our study miR-30c-5p is increased in cases of endometriosis, whereas in different types of cancer this miRNA is downregulated. Based on the experiments we have performed, we cannot explain what is the difference between the two pathologies linked to this miRNA. However, one of the hypotheses we suggest is based on the relationship between this miRNA and the TP53 gene, a gene considered to be a "tumor suppressor". In the study by Lin, S., et al. (2019) it is observed that low levels of miR-30c-5p are associated with increased mutations in the TP53 gene. Consequently, if this gene appears mutated it cannot exert its function as a tumor suppressor and consequently the tumor increases its aggressiveness. We have rewritten this part in the discussion to try to clarify our hypothesis. In addition, we have discussed the fact that miR-30c-5p is increased in endometriosis and decreased in cases of cancer such as ovarian or breast cancer, could be beneficial for a differential diagnosis between both pathologies.

See: Page 12, Discussion section, lines 324-327, 333-338, 346-351.

Reviewer 2 Report

Comments and Suggestions for Authors

1)  The authors proposed miR-30c-5p as a potential non-invasive biomarker of endometriosis, by analyzing the serum/plasma samples. Should not it be called minimal invasive not non-invasive? If saliva or urine had been the  biofluid, it could have been called non-invasive?

2)        Authors mentioned in line 44 that “One of the main reasons why CA-125 fails as a diagnostic method is that its blood levels fluctuate across the menstrual cycle” Please comment on the status of miR-30c-5p microRNA across the menstrual cycle.

3)        Please discuss the reproducibility of miR-30c-5p microRNA which has been mentioned as one of the main limitations of existing studies.

4)     The abbreviations need to be defined at their first application in the text such as ASRM And when an abbreviation was defined one time, it is not needed to define it again in the text.

Author Response

January 30, 2024

Dear Reviewer:

Thank you very much for taking the time to review our article entitled "Identification of miR-30c-5p microRNA in serum as a candidate biomarker to diagnose endometriosis". We appreciate the comments and suggestions of reviewers, as they have allowed us to improve the article and give more weight, credibility and depth to our results. Please find the detailed responses below and the corresponding revisions/corrections highlighted/in track changes in the re-submitted files. The line numbers detailed in the responses to the reviewers correspond to the manuscript titled "ijms-2845933-revised" where the modifications are marked. The final clean manuscript has also been uploaded to the platform.

Comment 1:

  • The authors proposed miR-30c-5p as a potential non-invasive biomarker of endometriosis, by analyzing the serum/plasma samples. Should not it be called minimal invasive not non-invasive? If saliva or urine had been the biofluid, it could have been called non-invasive?

Response 1:

Thank you for pointing this out. We agree with this comment because blood collection is a minimally invasive technique. Therefore, we have modified the term "non-invasive" to "minimally-invasive" when the text refers to serum samples.

See: Page 1, Abstract section, lines 14, 17, 27; Page 1, Introduction section, line 38; Page 2, Introduction section, line 58, 60, 72. Page 15, Conclusions section, lines 517-518.

Comment 2:

  • Authors mentioned in line 44 that “One of the main reasons why CA-125 fails as a diagnostic method is that its blood levels fluctuate across the menstrual cycle” Please comment on the status of miR-30c-5p microRNA across the menstrual cycle.

Response 2:

The studies by Cosar, E., et al. (2016) and Moustafa, S., et al. (2020) focused on the serum miRNA-typed biomarkers search for endometriosis did not observe differences in the expression levels of these molecules based on the phase of menstrual cycle (secretory vs proliferative). We agree with the reviewer. Based on the previous mentioned studies, we have analyzed the levels of our miRNA (miR-30c-5p) in the control group of healthy women during the proliferative and secretory phases of the cycle. Statistical analysis has also been performed with women in whom the phase of the menstrual cycle during which the blood sample was obtained was unknown. These analyses have been plotted in Figure A1. No differences in the level of miR-30c-5p have been detected in the different phases of the menstrual cycle. Results and Discussion sections have been modified to include this point.

See: Page 5, Results section, lines 120-123; Page 11, Discussion section, lines 264-268; Page 15, Materials and methods sections, lines 508-510; Page 17, Figure A1.

Comment 3:

  • Please discuss the reproducibility of miR-30c-5p microRNA which has been mentioned as one of the main limitations of existing studies.

Response 3:

Our study with respect to previous studies that have suggested different miRNAs as biomarkers of endometriosis is characterized by the following points:

  • The levels of each candidate miRNA during validation have been normalized using endogenous miRNAs with stable expression in the case and control groups, previously tested.
  • Quality controls have been included in the techniques performed.
  • The hemolysis variable have been controlled by two different methods.

Therefore, the use of miR-30c-5p as a biomarker of endometriosis is a reproducible technique in other laboratories if the previous specific points are carried out. Taking into account the reviewer's comments, we have detailed its reproducibility in the Discussion section to emphasize it.

See: Page 13, Discussion section, lines 373-378.

Comment 4:

  • The abbreviations need to be defined at their first application in the text such as ASRM and when an abbreviation was defined one time, it is not needed to define it again in the text.

Response 4:

We agree. We have, accordingly, revised the text to define each abbreviation the first time it is mentioned.

See: Page 1, Abstract section, lines 23-24; Page 2, Results section, lines 82, 83; Page 5, Results section, lines 109, 118, 119, 130; Page 7, Results section, line 175; Page 13, Materials and methods section, lines 384, 388-389; Page 14, Materials and methods section, lines 425-426; Page 15, Materials and methods section, lines 488, 503-504.

Reviewer 3 Report

Comments and Suggestions for Authors

Authors selected miRNA-30c-5p as potential serum biomarker for endometriosis. This paper is of interest and concerns important female reproductive problem. Several comments are addressed to Authors in order to clarify study findings:

1. The one of major limitation of this study is analyzing quite small miRNAs profile, and many others could be missed as biomarkers. However, this fact is not a basis to disqualify paper, but this fact regards to be disussed.

2. Table 1 should consist of numbers and percentages. Moreover, any additional information can be provided, such as presence of other diseases, laboratory markers that can affect the miRNSA expression.

3. Most of patients is ASRM stage III and IV, thus it needs to be underlined that selected miRNA is biomarker of mild of severe grade of endometriosis.

4. Did authors find any of information regarding changes in miRNA level during different stage of female monthy sexual cycle?

5. If authors did KEGG analysis, also GO can be presented.

6. Figure 1A please transfer to supplementary data.

Author Response

January 30, 2024

Dear Reviewer:

Thank you very much for taking the time to review our article entitled "Identification of miR-30c-5p microRNA in serum as a candidate biomarker to diagnose endometriosis". We appreciate the comments and suggestions of reviewers, as they have allowed us to improve the article and give more weight, credibility and depth to our results. Please find the detailed responses below and the corresponding revisions/corrections highlighted/in track changes in the re-submitted files. The line numbers detailed in the responses to the reviewers correspond to the manuscript titled "ijms-2845933-revised" where the modifications are marked. The final clean manuscript has also been uploaded to the platform.

Comment 1:

  • The one of major limitation of this study is analyzing quite small miRNAs profile, and many others could be missed as biomarkers. However, this fact is not a basis to disqualify paper, but this fact regards to be discussed.

Response 1:

We agree with this comment because there are different techniques to perform preliminary analysis, such as Next-Generation Sequencing (NGS). For example, Papari, E., et al. (2020) used NGS to identify differentially expressed plasma candidate miRNAs in women with endometriosis compared with control women. However, we used the QIAGEN serum/plasma-specific miRNA Focus PCR Panels. We selected this technique for preliminary analysis because, based on the studies published to date on miRNAs as biomarkers of endometriosis, the vast majority of suggested candidate miRNAs were collected in this panel. Subsequently, validation of candidate miRNAs was performed by quantitative PCR. This technique is the best tool for miRNA analysis, as it is fast, simple and inexpensive, and provides a sensitive analysis despite starting from low amounts of RNA.

We have modified the Discussion section to emphasize that there are other alternatives for preliminary study.

See: Page 10, Discussion section, lines 255-261.

Comment 2:

  • Table 1 should consist of numbers and percentages. Moreover, any additional information can be provided, such as presence of other diseases, laboratory markers that can affect the miRNSA expression.

Response 2:

We appreciate the reviewer's suggestion. Therefore, we have included the following information on the study subjects Table 1: Pain symptoms, Infertility, Tumoral markers, Comorbidities. These sections have been selected based on tables of patient demographic and clinical characteristics from previous studies on miRNAs and endometriosis performed on serum, plasma and saliva samples: Cho, S., et al. (2015); Dabi, Y., et al. (2023); Maged, A. M., et al. (2018); Moustafa, S., et al. (2020); Papari, E., et al. (2020).

See: Page 3, Table 1.

Comment 3:

  • Most of patients is ASRM stage III and IV, thus it needs to be underlined that selected miRNA is biomarker of mild of severe grade of endometriosis.

Response 3:

Thank you for pointing this out. We agree with this comment. Therefore, we have emphasized that miR-30c-5p would be a good biomarker for mild or severe stages of endometriosis.

See: Page 13, Discussion section, line 374; Page 15, Conclusions section, line 519.

Comment 4:

  • Did authors find any of information regarding changes in miRNA level during different stage of female monthly sexual cycle?

Response 4:

In the literature studies by Cosar, E., et al. (2016) and Moustafa, S., et al. (2020) found no changes in serum miRNA levels during different phases of the menstrual cycle in healthy women. We have analyzed whether our miRNA candidate biomarker of endometriosis, miR-30c-5p, fluctuates depending on the phase of the cycle. For this purpose, we compared the levels of this miRNA in the control group of healthy women. No statistically significant differences in miRNA miR-30c-5p levels were seen between the proliferative and secretory phases, nor in cases where the phase of the cycle is unknown. The results are illustrated in Figure A1.

See: Page 5, Results section, lines 120-123; Page 11, Discussion section, lines 264-268; Page 15, Materials and methods sections, lines 508-510; Page 17, Figure A1.

Comment 5:

  • If authors did KEGG analysis, also GO can be presented.

Response 5:

In the manuscript Figure 4a and the corresponding results section, the gene ontology biological processes were shown. To satisfy the reviewer's requirements, specific mentions to Gene Ontology analysis and specific results related to GO cellular components and GO molecular functions have been incorporated.

See: Page 7, Results section, lines 173-176, 177-182; Page 9, Figure 4 and lines 200-206; Page 15, Materials and methods section, lines 486-487, 489-490.

Comment 6:

  • Figure 1A please transfer to supplementary data.

Response 6:

Figure 1A has been transferred to supplementary material 1. The figure caption has been added to the manuscript in the "Supplementary materials" section.

See: Page 16, Supplementary Materials section, lines 527, 533-546.

Round 2

Reviewer 3 Report

Comments and Suggestions for Authors

Authors responded to all addressed comments accordingly. I have no additional questions.